# Field Study and Chemical Analysis of Plant Waste in the Fez-Meknes Region, Morocco

Ahmed Bendaoud [1,*], Amal Lahkimi [1], Mohammed Kara [2,*], Tarik Moubchir [3], Amine Assouguem [4,*], Abdelkhalek Belkhiri [5], Aimad Allali [6], Anouar Hmamou [1], Rafa Almeer [7], Amany A. Sayed [8], Ilaria Peluso [9] and Noureddine Eloutassi [1]

1   Laboratory of Engineering, Molecular Organometallic Materials and Environment, Faculty of Sciences Dhar El Mehraz, Sidi Mohamed Ben Abdallah University, Fez 30000, Morocco; amal.lahkimi@usmba.ac.ma (A.L.); anouar.hmamou@usmba.ac.ma (A.H.); noureddine.eloutassi@usmba.ac.ma (N.E.)
2   Laboratory of Biotechnology, Conservation and Valorisation of Naturals Resources (LBCVNR), Faculty of Sciences Dhar El Mehraz, University Sidi Mohamed Ben Abdallah, Fez 30000, Morocco
3   Polyvalent Laboratory in Research and Development, Department of Biology, Poly Disciplinary Faculty, Sultan Moulay Slimane University, Beni-Mellal 23000, Morocco; MOUBCHIR.TARIK.fpb21@usms.ac.ma
4   Laboratory of Functional Ecology and Environment, Faculty of Sciences and Technology, Sidi Mohamed Ben Abdellah University, Imouzzer Street, Fez 30000, Morocco
5   Laboratory of Bioactives-Health and Environment, Faculty of Sciences Meknes, Meknes 50000, Morocco; abdelkhalek.belkhiri@gmail.com
6   Laboratory of Plant, Animal and Agro-Industry Productions, Faculty of Sciences, University of Ibn Tofail (ITU), Kenitra 14000, Morocco; aimad.allali@usmba.ac.ma
7   Department of Zoology, College of Science, King Saud University, P.O. Box 2455, Riyadh 11451, Saudi Arabia; ralmeer@ksu.edu.sa
8   Zoology Department, Faculty of Science, Cairo University, Giza 12613, Egypt; amanyasayed@sci.cu.edu.eg
9   Research Centre for Food and Nutrition, Council for Agricultural Research and Economics (CREA-AN), 00178 Rome, Italy; i.peluso@tiscali.it
*   Correspondence: ahmedbendaoud@gmail.com (A.B.); mohammed.kara@usmba.ac.ma (M.K.); assougam@gmail.com (A.A.)

**Abstract:** Throughout the entire world, the biomass plant remains an important source of renewable energy. However, in Morocco, the energy recovery of this biomass is little or badly exploited compared to other solar, hydraulic, and wind resources. The aim of this study is to know the extent to which Moroccan companies are involved in the valorization of green waste and to identify among the latter those that have great energy and industrial value. The field investigation was carried out with the use of a questionnaire to different sectors of activity. The chemical analyses of the waste samples were carried out by different methods: Van Soest to investigate the fiber content, dinitrosalicylic acid and phenol-sulfuric acid to determine sugars, while the Folin–Ciocalteu method was employed for the determination of phenolic compounds. These are the ASTM standard methods to determine elemental, proximate composition, and calorific value (CV). The results of this survey showed that solid vegetable waste is diverse and represents 68.4% of the total green waste, of which 98% is not treated. Moreover, the chemical analysis displayed that forestry waste (FW), extracted parts wastes (EPW), and unused parts wastes (UPW) of medicinal and aromatic plants have high contents of cellulose (respectively 34.75, 48.44, and 54.19%) and hemicelluloses (28.44, 27.19 and 28.50%) and containing low amounts of lignin and phenolic compounds compared to olive waste (OW), olive pomace (OP), and household waste (HW). Almost all biomass wastes, except HW, have a low moisture (<12%), ash content less than 5.1%, a significant percentage of C and H, and CV between 14.5 and 21.6 MJ/Kg. The PCA analysis showed a discrepancy in terms of components between the set formed by FW, UPW, and EPW with other solid waste. In conclusion, FW, UPW, and EPW, specially can be potentially energetic biomass and valorized together in the form of a mixture.

**Keywords:** biomass; plant waste; renewable energy; lignocellulosic components; biomass resources

## 1. Introduction

Bioenergy production throughout the world has developed considerably within the past few decades; in 2020, it reached 127 GW with a rate of development of more than 30%. However, this rate is still very modest and insufficient because it represents only almost 5% of the total resources of renewable energy [1].

Lignocellulosic biomass is considered as an important source of green energy, and it is also considered as a component of industrial material [2,3] of cellulose, hemicellulose, and lignin, which are major components of this biomass that represent more than 70% of its total composition [4,5].

The valorization of lignocellulosic biomass components represents an effective solution, for the simple reason that it plays a very significant role in the growing energy and compensates biomaterial needs of the population, and on the other hand to reduce greenhouse gas emissions due to the abusive use of fossil oil resources [6,7].

It is widely known that Morocco has a very significant number of natural resources, particularly at the level of forestry residues. These residues come as a result of the exploitation of more than 30,000 ha/year [8,9]. In addition, these residues may result from the wastes of medicinal and aromatic plants ranging from medicinal, cosmetic, and other sectors that Morocco is famous for [10]. In this respect, it is worthy to mention that we have some other extra resources from which we can obtain the residues, such as, for instance, olive pomace and the olive harvest, which is regarded as one of the most vitally important sources that we can depend upon because Morocco produces almost 500,000 tons of olive annually [11], without forgetting to mention the organic household wastes resulting from the daily normal consumption.

Despite the existence of a significant amount of waste resulting from the use of these natural resources, their exploitation is still almost strictly traditional. For instance, forest wastes such as "wood" are used for a variety of purposes such as the cooking of bread and meals in the oven, heating in the season of winter, or traditional Moroccan baths [12]. We can also mention the medicinal and aromatic plants, which have several ways to be used, especially in treating diseases, while their wastes are left untreated or used as compost [13,14]. Furthermore, we also have the olive pomace that is used as fuel and natural compost [15]; however, the use of these wastes is still quite limited in sustainable development. Therefore, we can safely say that Morocco, which is regarded as one of the users and consumers of these natural resources, has produced only about 0.1% of biogas of the total renewable energy in 2020, which reached 3500 MW [16]. Despite this weak production, according to the World Energy Organization, Morocco is considered one of the initiators to invest in bioenergy compared to other north African countries such as Algeria, Tunisia, Libya, and Mauritania, which are characterized by a general weakness in the production of renewable energies and a lack of bioenergy. On the other hand, if we compare Morocco with other countries on the continent, such as Ethiopia which produces 290 MW and South Africa which produces 265 MW, it needs more efforts in this field.

The main objective of this study that we have at hand is to carry out a field investigation in the Fez-Meknes region on the types of vegetable waste, the state of their treatment, the obstacles that prevent investment in them, as well as their recycling by the producing organisms. An additional aim of this study is to determine, among these rejected wastes, the vegetable biomass that can be energetically valorized by analyzing their chemical compositions in fibers, sugars, phenolic compounds, elemental, and proximate composition.

## 2. Materials and Methods

### 2.1. Sampling

#### 2.1.1. Study Area

This study was carried out in the Fez-Meknes region, which is considered as one of the twelve largest regions in the kingdom of Morocco that comprises nine provinces, which can be classified as follows: Fez, Meknes, Ifrane, Taza, Taounate, Sefrou, El Hajeb, Boulemane, and last but not least, Moulay Yacoub. This particular region covers an area of

40,075 km$^2$, or 5.7% of the general national territory. The population of the region of Fez-Meknes has reached 4,236,892 inhabitants, which makes it 13% of the national population; however, 60.52% of this population live in urban areas while the rest of them live in the surrounding countryside. It is worthy to mention that the region has important natural resources especially at the level of forests, which the region is renowned for, and that covers an area of 1,446,160 square hectares, which is 16% of the national area. Therefore, we can safely say that 80% of these forests are located in the provinces of Boulemane and Taza. In this respect, the fertile agricultural land represents 1,335,639 hectares; that is to say, it shapes 15% of the national useful agricultural area. It is widely known that the economy of the region depends mainly and exclusively upon agricultural activities. In fact, agriculture contributes with 21% of the national production of cereals and also contributes with 6% by some other extra industrial activities. These activities are represented by the agro-food, textile, and leather industries that are 80% concentrated in the provinces of Fez and Meknes [17].

### 2.1.2. Respondents

As a matter of fact, the survey of this research paper was carried out by means of a questionnaire that was devised and distributed in 100 establishments and industries in the Fez-Meknes region ranging from the private sectors, public sectors, and cooperative associations as follows:

- Wood industries.
- Olive oil mills.
- Medicinal and aromatic plant cooperatives.
- Regional Council Fez-Meknes.
- Delegation of water and forests of the region Fez-Meknes.

### 2.1.3. Plant Material

The type of plant wastes that were selected for this study can be classified as the following:

- The wastes of the most used medicinal plants in Moroccan traditions before and after their extraction (*Thymus vulgaris*, *Rosmarinus officinalis*, *Origanum vulgare*, *Mentha pulegium Mentha spicata*, *Mentha piperita*, *Artemisia absinthium*, *Salvia Officinalis*) [18],
- Vegetable household wastes (vegetables and fruits),
- Olive tree residues during the harvest (branches and leaves),
- Olive pomace and forest residues (leaves, branches, cones, seeds, sawdust).

These samples were collected from various and different industries, associations, and cooperatives in the Fez-Meknes region. Then, these collected samples were dried at a temperature of 55 °C for a period of time 72 h until they reached a stable weight. After, they were grinded to reduce their size and, therefore, to obtain a uniform size biomass of 1 mm.

### 2.2. Fiber Dosage

Cellulose, hemicellulose, and lignin were determined by Van Soest's method [19,20]. The results are expressed in % of the initial dry mass:

- Cellulose = (ADF − ADL)/initial dry mass.
- Hemicelluloses = (NDF − ADF)/initial dry mass.
- Lignin = ADL/initial dry mass.

### 2.3. Dosage of Sugars and Total Phenolic Compounds

The concentration of reducing sugars is determined by the dinitrosalicylic acid method (DNS) [21]. Briefly, in a 20 mL test tube containing 3 mL of Miller's reagent, 2 mL of test solution is added. After heating the solution for a period of time of 15 min at a temperature of 100 °C, the solution was left to cool at a normal temperature. Thus, the absorbance is measured at 640 nm. The total sugars are measured by the phenol-sulfuric acid method by measuring the optical density at 490 nm [22].

Additionally, in this research the total phenolic compounds were analyzed by the Folin–Ciocalteu method using gallic acid as a standard, as it is described by Singleton and Rossi [23].

*2.4. Determination of Elemental, Proximate Compositions, and Caloric Value of Plants Wastes*

Proximate testing includes moisture content, ash, and volatile matter. This testing is conducted by combusting the biomass samples in a high temperature furnace:

- The percentage of Moisture content was carried out using the ASTM E871 standard with the following equation:

$$Moisture\% = (A - B)/A$$

where A is the mass of the sample used and B is the mass of the sample after heating in 105 °C.

- The ash content percentage in biomass wastes was performed using the ASTM E830. standard with the following equation:

$$Ash\ content\% = C/A \times 100$$

where C is the weight of sample after heating 575 °C.

- The percentage content of volatile matter contained in biomass wastes was conducted using the ASTM E872 standard with the following equation:

$$Volatile\ matter\% = (B - D)/A$$

where D is the weight of sample after heating.

The percentage composition of the fixed carbon (FC) was calculated from the difference as follows:

$$FC\% = 100 - (Ash\% + VM\%)$$

The calorific value is obtained by using a bomb calorimeter. The calorific value obtained by bomb based on the ASTM D240 standard.

The percentage of elemental composition contained in biomass wastes measured by elemental analyzer was used to determine the contents of carbon (C), hydrogen (H), nitrogen (N), and sulfur (S) based on the ASTM E777, E778, E775 standard.

The content of oxygen (O) was determined using the formula as follows:

$$O(\%) = 100 - (\%C + \%H + \%N + \%S)$$

*2.5. Statistical Analysis*

The statistical analyses were processed with the use of the software (IBM SPSS statistical version 27, University of New Haven; West Haven, CT, USA). The comparison between the different measured parameters was performed by the one-factor ANOVA test followed by Tukey's multiple comparison with $p < 0.05$. Principal component analysis (PCA), while the correlation was determined between the studied variables.

## 3. Results

*3.1. Survey Analysis*

3.1.1. General Profile of the Respondents

Table 1 represents the general profile of the respondents. A total of 63% of the units that were visited belong to the private sector while 35% belong to a variety of cooperatives. It is also worthy to mention that almost half of the managers are between 45 and 55 years old; however, more than two thirds are men while more than 65% of those managers' have a level that do not exceed the primary level with only 22.4% have a university education.

Thus, we can say that almost a third of those previously mentioned companies were created in the last decade.

**Table 1.** General information of respondents.

| Variable | Sub-Group | Number | Percentage% |
|---|---|---|---|
| Age | 25–35 | 3 | 3.0 |
| | 35–45 | 15 | 15.0 |
| | 45–55 | 48 | 48.0 |
| | Over 55 | 34 | 34.0 |
| Sex | Man | 72 | 72.0 |
| | Woman | 28 | 28.0 |
| School level | Nonet | 27 | 27.6 |
| | Primary | 38 | 38.8 |
| | College | 6 | 6.1 |
| | Secondary | 5 | 5.1 |
| | University | 22 | 22.4 |
| Sectors of production units | Private | 63 | 63.0 |
| | Public | 2 | 2.0 |
| | Social economy | 35 | 35.0 |
| Year of creation | Before 2000 | 29 | 29.6 |
| | 2000–2010 | 34 | 34.7 |
| | After 2010 | 35 | 35.7 |

### 3.1.2. Types of Solid Plant Waste

Table 2 displays the major types of plant waste discharged by the different production units. We can notice that solid waste represents 68.4% that have various kinds, such as, for instance, stems, roots, seeds, and leaves. According to the data collected, olive oil mills generate important quantities of olive pomace, which reaches approximately an average of 2854.84 ton/year, while wood industries produce 13.59 ton/year, mainly made of sawdust. As for the public institutions surveyed, the regional council of Fez-Meknes and Delegation of Water and Forests of the region Fez-Meknes, it was revealed that the producers do not benefit from vegetable waste because the first does not select household waste to benefit separately, while for the second they are not treated and used traditionally.

**Table 2.** Types of plant waste discharged by generating organizations.

| Production Units | Number | Fraction Used | Solid Plant Waste | Quantity of Waste (Average in Ton/Year) |
|---|---|---|---|---|
| Wood industries | 39 | Wood | Sawdust, planing shavings | 13.59 |
| Medicinal and aromatic plant cooperatives | 21 | Leaves, Biomass Blend | Stems, roots, seeds | - |
| Olive oil mills | 31 | Seeds and fruits | Olive pomace | 2854.84 |
| Delegation of water and forests of the region Fez-Meknes | 1 | - | Leaves, branches, cones, seeds, sawdust | - |
| Regional Council Fez-Meknes | 1 | - | Household fruit and vegetable waste | - |

### 3.1.3. Treatment of Vegetable Waste

The major remark that we can deduce from Figure 1 is that 62.24% of the respondents perceive that waste treatment is not important for them, while 37.76% confirmed that it is

indeed economically important; however, the totality ignores the environmental importance of waste treatment. Additionally, we can notice that 58.16% of the production units do not have a dumping area while 98% of the waste is not treated by these units, which forces them to be sold raw to other actors. These products that are heavily sold are olive pomace and sawdust. Whereas the products that are used by the units traditionally are extracted from medicinal and aromatic plants, some other products can be left unused as forestry waste.

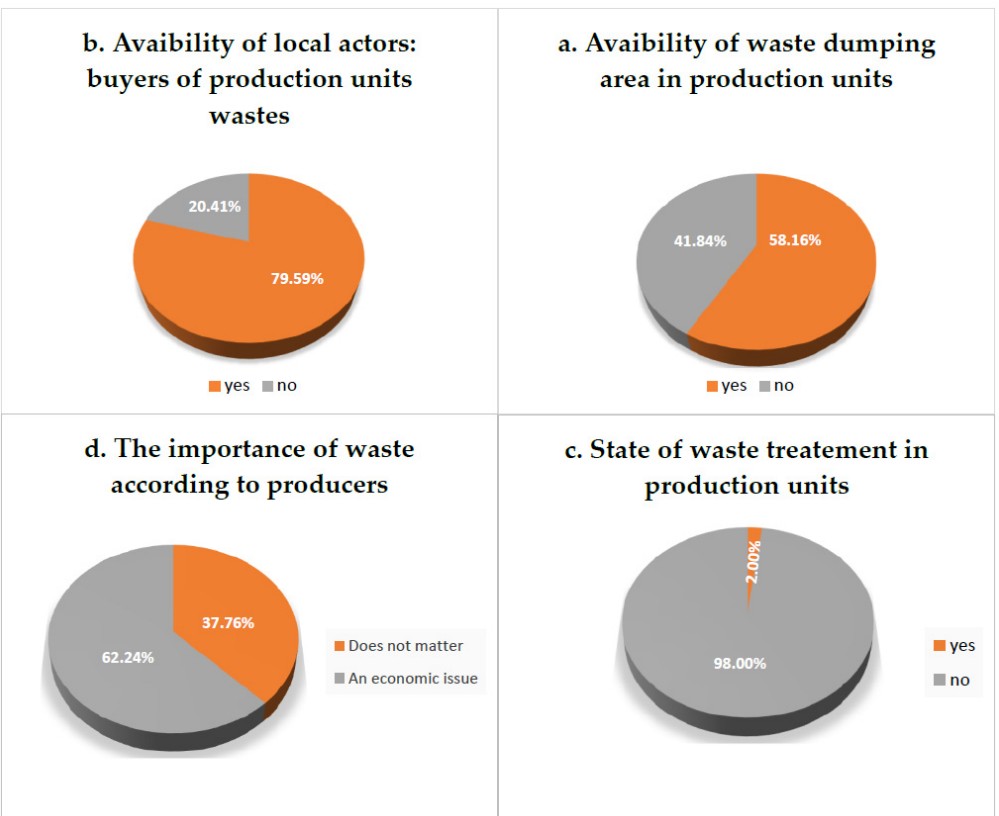

**Figure 1.** Treatment of vegetable waste and its importance.

3.1.4. Obstacles to Plant Waste Treatment

The circular economy is regarded as a new economic model used recently instead of the linear economy as a basic principle of sustainable development and to react against the depletion of natural resources; however, unfortunately, 81.5% of the respondents do not have a clear idea about the circular economy. The vast majority of actors in these various sectors of activity consider that the lack of means and lower profit are the main obstacles that lead to the non-treatment of this waste.

*3.2. Fiber Content*

The analysis of the results in the charts of cellulose, hemicellulose, and lignin contents are presented in Table 3. According to these illustrated results, cellulose is the most important component of all the studied sample; that is to say, it represents high contents of cellulose at values of 34%, 48%, and 54% of the dry mass in FW, OP, and EPW waste, respectively. The hemicellulose content is also vitally important; however, it is less answered compared to cellulose with an average rate of 17%, 24%, 27%, and 28% in OP, OW, EPW, FW, and UPW wastes, respectively, while lignin represents only about 16% of dry matter in most of the samples studied, except in HW where its ratio is very low (2.35%).

The analysis of variance (ANOVA test) for the cellulose, hemicelluloses, and lignin contents of the plant waste are significantly different ($p < 0.05$) for each measured parameter (Table 3). The Tukey multiple comparison that was carried out between the means for each measured parameter (cellulose, hemicelluloses, lignin,) displayed that:

- The subsets: "HW", "OW", "FW, OP", "EPW", "UPW" have a significant difference at the level of cellulose contents.
- The subsets: "HW", "OW, FW, OP, EPW, UPW" contain significant difference at the level of lignin.
- For each of the biomass's OP, OW, and HW, their hemicellulose contents are significantly different, while the biomasses FW, UPW, and EPW do not show a significant difference between them.
- The rate of the soluble fraction is quite dissimilar in all the samples studied, except between OP and OW where there is no significant difference.

**Table 3.** Cellulose, hemicellulose, and lignin contents in various plant wastes (% mass dry matter).

| | Samples | | | | | |
|---|---|---|---|---|---|---|
| | **FW** | **UPW** | **EPW** | **OP** | **OW** | **HW** |
| % Cellulose | 34.75 ± 1.2 [a] | 54.19 ± 0.50 [b] | 48.44 ± 0.60 [c] | 34.00 ± 0.50 [a] | 27.81 ± 1.5 [d] | 7.35 ± 0.70 [e] |
| % Hemicellulose | 28.44 ± 1.5 [a] | 28.50 ± 0.60 [a] | 27.19 ± 0.70 [a,c] | 17.13 ± 0.60 [b] | 24.69 ± 1.20 [c] | 4.29 ± 0.25 [d] |
| % lignin | 17.63 ± 1.3 [a] | 16.81 ± 0.40 [a] | 16.63 ± 1.5 [a] | 16.20 ± 1.7 [a] | 16.44 ± 0.50 [a] | 2.35 ± 0.40 [b] |
| % Soluble Fraction | 19.19 ± 0.50 [a] | 50 ± 0.25 [b] | 7.75 ± 0.80 [c] | 32.00 ± 1.1 [d] | 31.06 ± 0.50 [d] | 86.00 ± 1.50 [e] |

FW—forestry waste; UPW—unused parts of medicinal and aromatic plant waste; EPW—extracted parts waste of medicinal and aromatic plants; OP—olive pomace; OW—olive waste; HW—household waste; [a], [b], [c], [d], and [e]—values with a significant difference ($p < 0.05$). Data reported as mean ± standard deviation from three replicate determinations.

### 3.3. Content of Sugars and Phenolic Compounds

The wastes that contain high concentration of sugars and low content of phenolic compounds are UPW, EPW, and FW (Figure 2), whose total sugar concentrations are 22.85, 19.34, and 18.45 g/L and phenolic compounds 14.59, 16.45, and 14.45 g/L, respectively. The OP and OW wastes are rich in phenolic compounds and contain less sugars.

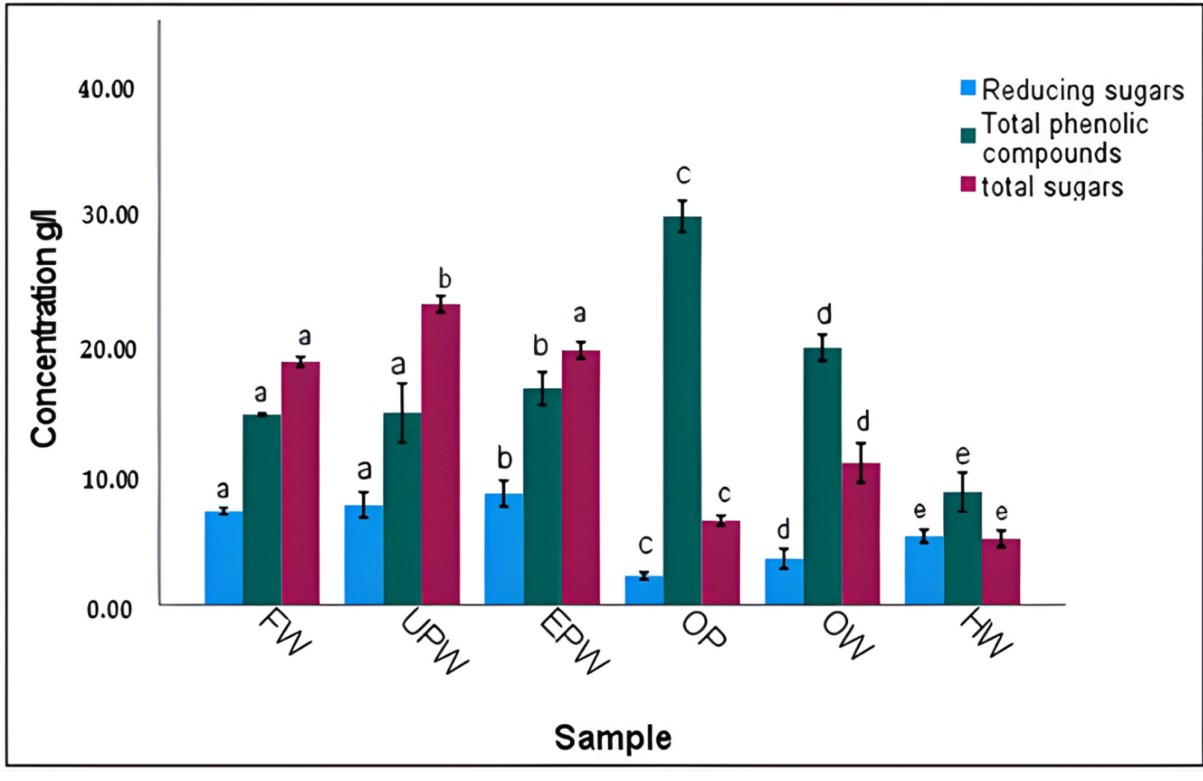

**Figure 2.** Sugar and phenolic compound contents (g/L) a, b, c, d, and e—values with a significant difference ($p < 0.05$).

### 3.4. Correlation between the Different Parameters Measured

The different parameters that have been studied and investigated revealed that the Pearson correlation coefficient on the one hand shows a positive correlation between several parameters (Table 4), in particular between cellulose-hemicelluloses (r = 0.846), cellulose-lignin (r = 0.792), hemicelluloses-lignin (r = 0.978), and cellulose-total sugars (r = 0.844). On the other hand, the soluble fraction is strongly and negatively correlated with total sugars, lignin, hemicellulose, and cellulose (respectively, r = −0.842, r = −0.907, r = −0.937, and r = −0.964) and total phenolic compounds with reducing sugars (r= −0.623).

**Table 4.** Pearson correlation coefficients between different parameters.

| | Cellulose | Hemicellulose | Lignin | Soluble Fraction | Total Phenolic Compounds | Reducing Sugars | Total Sugars |
|---|---|---|---|---|---|---|---|
| Cellulose | 1.000 | 0.846 | 0.792 | −0.964 | 0.258 | 0.516 | 0.844 |
| Hemicellulose | | 1.000 | 0.978 | −0.937 | 0.551 | 0.175 | 0.643 |
| Lignin | | | 1.000 | −0.907 | 0.546 | 0.137 | 0.611 |
| Soluble fraction | | | | 1.000 | −0.311 | −0.451 | −0.842 |
| Total phenolic compounds | | | | | 1.000 | −0.623 | −0.242 |
| Reducing sugars | | | | | | 1.000 | 0.823 |
| Total sugars | | | | | | | 1.000 |

The obtained principal component analysis (PCA) shows that the values of the first two principal components represent 96.2% of the information of all the studied variables.

Based on the projection of the correlation circle (Figure 3) and the representation of the biomass distribution (Figure 4), we can notice that there is a positive contribution of cellulose, hemicellulose, and lignin on the first principal axis in positive correlation with FW, UPW, and EPW. However, we can also remark a reducing sugars and total polyphenols contributed on the second main axis in correlation with OW and OP.

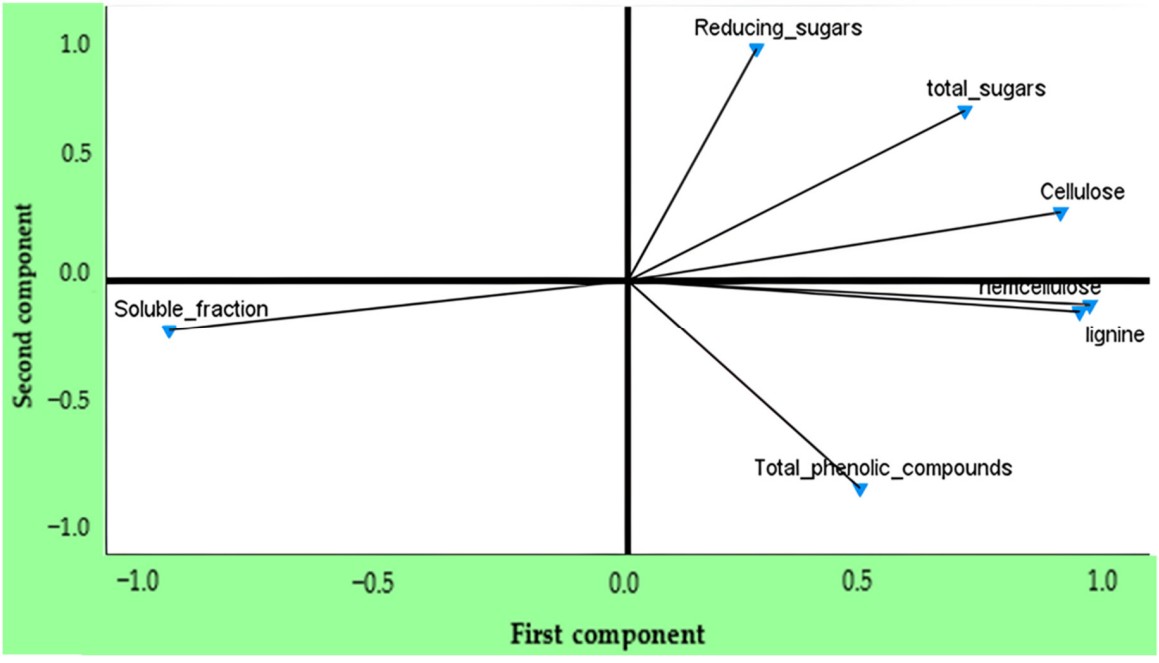

**Figure 3.** Principal component analysis of different studied parameters.

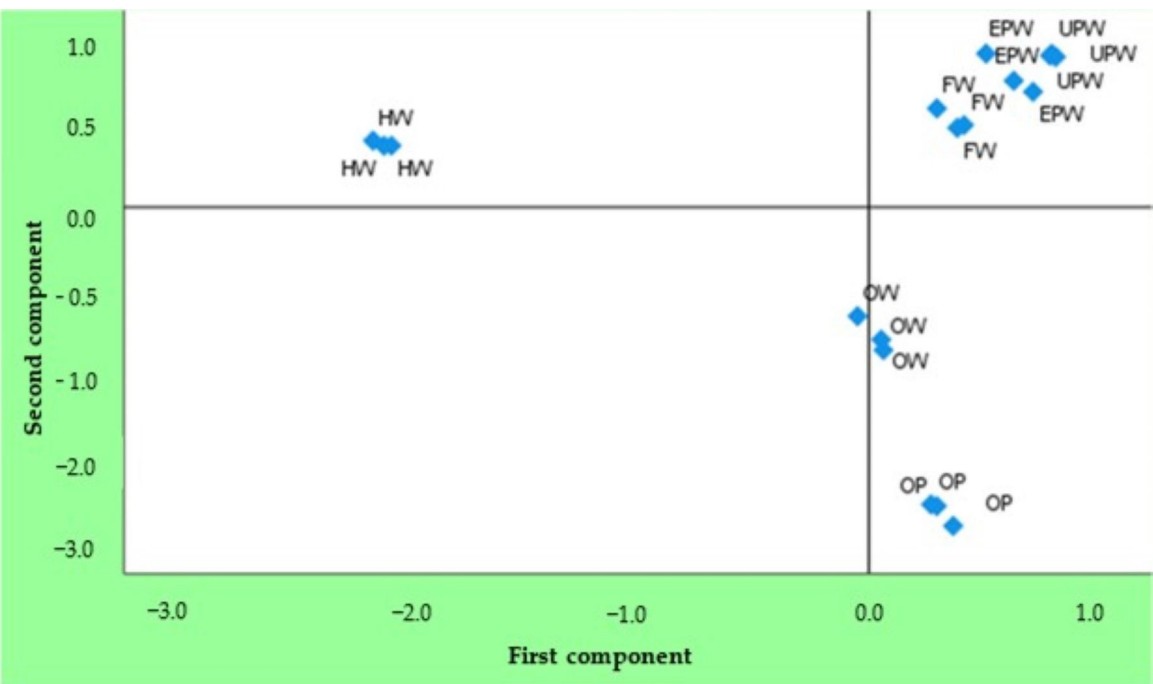

**Figure 4.** Representation of biomass distribution.

### 3.5. Elemental, Proximate Compositions and Caloric Value of Plant Waste

The characterization of different biomass wastes included proximate analyses, elemental composition, and calorific values can be seen in Table 5. A high moisture content of 11.7% was obtained for FW and does not exceed 8.2% for the other samples studied. The ash content is very low for FW in the order of 1.5% and elevated for HW, which represents 10.4%. The percentage of volatile matter ranges between 63.2 and 76.1%. FW and OP has a high carbon content with 51 and 54%, respectively, and of hydrogen content with 7.5 and 7.6%, respectively, compared to other biomass residues. Almost all the biomass wastes have a low nitrogen contents and negligible amount of sulfur except HW, which contain 3.7% of nitrogen and 0.4% of sulfur. The caloric value in dry basis of biomass wastes with 18.5–21.6 MJ/kg observed in FW, UPW, OP, and OW, indicate a significant energy content in comparison to other biomass residues, while the HW represent a lower percentage of 12.5 MJ/Kg.

**Table 5.** Elemental, proximate compositions, and caloric value of plants wastes.

| | Samples | | | | | |
|---|---|---|---|---|---|---|
| | **FW** | **UPW** | **EPW** | **OP** | **OW** | **HW** |
| Moisture (M) (wt.%) | 11.7 | 8.2 | 7.6 | 5.8 | 3.4 | 3.5 |
| Volatile matter (VM)% | 71 | 78.4 | 63.2 | 74 | 76.1 | 70.5 |
| Fixed carbon (FC)% | 15.5 | 9.6 | 24 | 16.5 | 18.2 | 15.4 |
| Ash (dry wt.%) | 1.5 | 2.7 | 5.1 | 4.5 | 2.2 | 10.4 |
| % Carbon (C) | 51 | 46.9 | 39.2 | 54 | 48.3 | 46.5 |
| % Oxygen(O) | 40.6 | 42.8 | 54.2 | 35.3 | 45 | 32 |
| % Hydrogen(H) | 7.5 | 6.7 | 5.3 | 7.6 | 5.9 | 6.8 |
| % Sulfur(S) | 0 | 0.3 | 0.09 | 0.08 | 0.1 | 0.4 |
| % Nitrogen(N) | 0.2 | 1.8 | 1.1 | 2.3 | 0.7 | 3.7 |
| Caloric value (MJ/Kg) | 19 | 19.8 | 14.5 | 21.6 | 18.5 | 12.5 |

## 4. Discussion

The various types of wastes discharged by the numerous sectors of activity illustrated that the producers do not benefit from it, simply because according to them it comes due to

the lack of means and lower profit. Despite the fact that there is great potential in available biomass as well as with the growing demand for energy, the energy recovery of biomass is developing on slow paces in the region Fez-Meknes. Therefore, we can safely say that it is generally limited to a few initiatives of public institutions such as the controlled landfill of Fez that is responsible for producing thermal energy from the combustion of household waste and the Wastewater Treatment Plant (WWTP of Fez) and which biologically treats organic biomass by producing biogas. The energy strategy, which is based mainly on renewable energy (RE), that was launched in 2009 has managed to cover 34.6% of the total electrical capacity [24]. However, until now, biomass has not been well or fully invested in this project; that it is to say, it has been limited mainly to other renewable resources. For this reason, Morocco has developed an action plan that aims and seeks to develop the energy value of biomass and produce between 17–25 TWh/year or 52% of the total electrical capacity by 2030. This plan is heavily based on three axes [25]:

Establishment of a legislative framework favorable to the use of biomass for energy purposes.

The durable transformation of organic wastes in resources to create a regional added value.

Guarantee resources and means for a sustainable and innovative development of biomass valorization.

The results of the chemical composition analyses showed and explained that the forest waste (WF), the waste of extracted parts (EPW), and unused parts (UPW) of medicinal and aromatic plants contain respectively high contents of cellulose (34.75, 48.44, and 54.19%), hemicellulose (28.44, 27.19, and 28.50%), and little lignin (17.63, 16.63, and 17.81%). These results obtained are almost similar to previous studies carried out on isolated samples as forest residues of plants *Eucalyptus globulus*, *Pinus pinaster*, or on part of the plant as *Pinus pinaster bark*, *Quercus suber L. bark*, their contents of cellulose (24–47.7%) and hemicellulose (15–20.59%) and lignin (13–33.2%) [26]. Other studies conducted on *Rosmarinus officinalis* [27] and the residues of *Mentha arvensis* [28] also showed comparable results respectively their contents of cellulose (48, 42.8%), hemicellulose (33, 28.2%) and lignin (17, 17.6%). The work, therefore, on a mixture of waste (stems, leaves, root...) will save the cost and time of sorting and have the raw material with significant quantities.

The high content of cellulose and hemicellulose in most of the analyzed wastes does not mean that they are easily exploitable because the lignocellulosic biomass usually shows low biogas production and biodegradability due to the recalcitrance of the lignocellulosic complex [29,30]. Thus, the negative correlation between phenolic compounds, total sugars, and reducing sugars (Table 3), confirms the inhibitory effect of phenolic compounds on the hydrolysis of polysaccharides into fermentable simple sugars [31]. Therefore, biomasses containing a low concentration of phenolic compounds and rich in sugars, allow to valorize them energetically at the lowest cost without resorting to an additional treatment.

Moreover, proximate composition and calorific value of various biomass wastes, are the key factors for thermo-conversion and bioconversion of biomass. Technically, the high amount of moisture (>50%) [32], the ash contents more than 8% [33] and low calorific value in a dry feedstock, limits the conversion pathway for producing biofuels, bioproducts, and biopower. According to the results of proximate and calorific value, the most vegetable wastes except HW can be considered a high raw materials energy efficient. In regard to the elemental composition, it is noticeable that C, H contents positively affected the heating value of biomass and, therefore, leads to a change in the energy content of the biomass [34,35]. Additionally, the high S content needs to be considered, as its combustion will increase the sulfur dioxide content in the air [36]. Among vegetable wastes analyzed, carbon and oxygen content percentages are higher, with low hydrogen and nitrogen percentage and no sulfur content in FW, UPW, EPW, and OP compared to OW and HW biomass wastes.

In fact, the problem of waste treatment by production units, according to the survey, cannot be attributed only to poor capacity and lower profit, but also to ignorance of their energy and industrial values. It became clear through the results of chemical analysis that they are raw materials rich in fibers, which can be burned, transformed into a fuel gas

through combustion, into a biogas through fermentation, or into a biomaterial. Therefore, this ignorance of the value of plant waste can be explained by the weak level of training of the owners and managers of these production units, which do not exceed the primary level and, thus, are unable to keep pace with progress in this context.

## 5. Conclusions

We can safely state that the vegetable waste that has been rejected by various and numerous sectors of activity in the Fes-Meknes region is little and poorly exploited and mostly untreated. Chemical analysis of the samples of these wastes reveals that the mixtures of each of the biomasses of forestry wastes and medicinal and aromatic plant wastes seem particularly interesting and can be important energy resources, because these forestry wastes and the medicinal and aromatic plants are rich in cellulose and hemicelluloses and contain less lignin and phenolic compounds compared to the other biomasses analyzed. Due to the fact that there is a lack of means to treat the rejected waste and the low income according to the actors, the work on a mixture of biomasses could be an effective solution to save the sorting stage and to benefit easily from it.

The valorization of lignocellulosic material from these plant wastes as a source of energy and biomaterials requires adequate pre-treatment that takes into consideration the cost, time, and method used to hydrolyze cellulose and hemicellulose into simple fermentable sugars. This could encourage actors to invest in this field and increase the contribution of plant biomass as a vitally important source of renewable energy in Morocco and in developing countries as a whole.

**Author Contributions:** Conceptualization, A.B. (Ahmed Bendaoud) and M.K.; methodology, A.B. (Ahmed Bendaoud), A.A. (Amine Assouguem), T.M. and A.B. (Abdelkhalek Belkhiri); software, M.K. and A.A. (Amine Assouguem), A.H. and A.A. (Aimad Allali); validation, N.E. and A.A.S.; formal analysis, A.B. (Ahmed Bendaoud) and M.K.; investigation, A.B. (Ahmed Bendaoud), A.A. (Amine Assouguem), A.A. (Aimad Allali), and N.E.; data curation, A.B. (Ahmed Bendaoud) and R.A.; writing—original draft preparation, A.B. (Ahmed Bendaoud), M.K., A.L. and A.A. (Amine Assouguem), and N.E.; writing—review and editing, A.B.(Ahmed Bendaoud), M.K., I.P. and N.E.; supervision, M.K., A.L. and N.E. All authors have read and agreed to the published version of the manuscript.

**Funding:** This research was funded by Researchers Supporting Project number (RSP-2021/96), King Saud University, Riyadh, Saudi Arabia.

**Institutional Review Board Statement:** Not applicable.

**Informed Consent Statement:** Not applicable.

**Data Availability Statement:** Not applicable.

**Acknowledgments:** The authors would like to extend their sincere appreciation to the Researchers Supporting Project number (RSP-2021/96), King Saud University, Riyadh, Saudi Arabia.

**Conflicts of Interest:** The authors declare no conflict of interest.

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
