# Peer review of "Field Study and Chemical Analysis of Plant Waste in the Fez-Meknes Region, Morocco"

_sustainability, doi:10.3390/su14106029_

Round 1

Reviewer 1 Report

The author investigated the origin of vegetable waste and analysed the chemical components of the waste. The recycling and reuse of waste material is an important issue in order to save the environment. In my opinion, the present manuscripts need to be revised thoroughly to make it informative to the reader.

Some of my comments are as follows.

  1. Abstract is too lengthy and not clear. Please modify it so that the essence of this work can be reflected in the abstract and readers find interest in it.
  2. Introduction needs to improve; the quality of the English language needs to improve and there is no consistency in writing.
  3. Table 2 there is no column title so it's difficult to understand what the authors want to explain. For e.g., what is the meaning of the Fraction used, 39,21,31,1 means?  Please make a clear and well explained table.
  4. Also explain the important information and suggestions obtained from table 1 and 2.
  5. There is no detailed explanation of figure 1, moreover, the picture quality needs to improve.
  6. I would like to suggest to authors to carefully revise the manuscript, so that readers can understand and get the essence of this work.

Author Response

Response to Reviewer 1 

It is a pleasure for our team to receive the recommendations for reviewing our manuscript.

We would like to express our gratitude for the time and expertise you invested in improving our manuscript and enhancing its quality for publication in Sustainability journal.

We have carefully considered all suggestions, we have introduced all recommendations, and we have adjusted the manuscript to fit your requirement.

We remain available for further comments and adjustments of our manuscript, as well as for additional explanations.

Please find below the itemized answers. 

The author investigated the origin of vegetable waste and analysed the chemical components of the waste. The recycling and reuse of waste material is an important issue in order to save the environment. In my opinion, the present manuscripts need to be revised thoroughly to make it informative to the reader.

Point 1: Abstract is too lengthy and not clear. Please modify it so that the essence of this work can be reflected in the abstract and readers find interest in it.

Response 1:

Thank you for this kind remark; it is interesting to reformulate it to be clear to the readers:

  • We have removed the following sentence:

“The energy from this biomass is storable and controllable according to needs”

  • we reformulate our objective for this study as follows :

“ The aim of this study is to know the extent to which Moroccan companies are involved in the valorization of green waste and to identify among the latter those that have great energy and industrial value”

  • we have added and reformulte this paragraph concerning materials and methods in the abstract:

“the chemical analyses of the waste samples were also carried out by different methods. Van Soest to investigate the fiber content, Dinitrosalicylic Acid and Phenol-Sulfuric Acid to determine sugars, while the Folin-Ciocalteau method was employed for the determination of phenolic compounds. Thus, the ASTM standard methods to determine elemental, proximate composition and calorific value (CV).”

  • We added the results about elemental, proximative composition and calorific value of biomass wastes as follows:

“Almost biomass wastes except HW, have a low moisture (<12%), ash content less than 5.1%, a significant percentage of C and H and CV between 14.5 and 21.6 MJ/Kg”

Point 2: Introduction needs to improve; the quality of the English language needs to improve and there is no consistency in writing.

Response 2:

Thank you for your comment; Indeed, the manuscript was reviewed by an Native English Speaker.

About the clearness of the introduction for the reader, we have reformulate it and added some information as follow :

“ Although this weak production, according to the World Energy Organization, Morocco is considered one of the initiators to invest in bioenergy compared to other north African countries such as Algeria, Tunisia, Libya and Mauritania, which are characterized by a general weakness in the production of renewable energies and a lack of bioenergy. On the other hand, if we compare Morocco with other countries on the continent, such as Ethiopia, which produces 290 MW and South Africa 265 MW; it needs more efforts in this field.”

Point 3: Table 2 there is no column title so it's difficult to understand what the authors want to explain. For e.g., what is the meaning of the Fraction used, 39,21,31,1 means?  Please make a clear and well explained table.

Response 3:

The column titles in Table 2 are corrected.

Point 4: Also explain the important information and suggestions obtained from table 1 and 2.

Response 4:

Indeed, concerning table 1, we have added information about the year of creation of the production units as follows:

Thus, we can say that almost a third of those previously mentioned companies were created in the last decade.”

Added in result section

About table 2, we have added the following explanation:

“As for the public institutions surveyed: the regional council of Fez-Meknes and Delegation of water and forests of the region Fez Meknes, it was revealed that the producers do not benefit from vegetable waste, because for the first does not select household waste to benefit separately, while for the second are no treated and used traditionally”.

Added in result section

Point 5: There is no detailed explanation of figure 1, moreover, the picture quality needs to improve.

Response 5:

Thank you for your notice. the picture is detailed and the quality was improved.

Point 6: I would like to suggest to authors to carefully revise the manuscript, so that readers can understand and get the essence of this work.

Response 6:

Indeed, with the assistance of your pertinent remarks, we have tried to process all your suggestions.

Reviewer 2 Report

General comments:

This manuscript focused on the field study and chemical analysis of plant waste in the Fez-Meknes region, Morocco. On the whole, it is a meaningful work and will provide significant reference for the application of plant waste. However, there are still some problems existing in this manuscript. I suggest that the authors revised their manuscript according to the following comments. After this, this manuscript can be considered by this journal.

Special comments: 

(1)The elemental/proximate compositions of plant waste must be added.

(2)The caloric value of plant waste must be added.

(3)The pyrolysis and incineration characteristics should also be considered.

Author Response

Response to Reviewer 2

It is a pleasure for our team to receive the recommendations for reviewing our manuscript.

We would like to express our gratitude for the time and expertise you invested in improving our manuscript and enhancing its quality for publication in Sustainability journal.

We have carefully considered all suggestions, we have introduced all recommendations, and we have adjusted the manuscript to fit your requirement

We remain available for further comments and adjustments of our manuscript, as well as for additional explanations.

Please find below the itemized answers. 

This manuscript focused on the field study and chemical analysis of plant waste in the Fez-Meknes region, Morocco. On the whole, it is a meaningful work and will provide significant reference for the application of plant waste. However, there are still some problems existing in this manuscript. I suggest that the authors revised their manuscript according to the following comments. After this, this manuscript can be considered by this journal.

Point 1: The elemental/proximate compositions of plant waste must be added.

Response 1:

Thanks for this exciting suggestion, please find the results of the analyses in the following table 5:

Table 5: Elemental, proximate compositions, and caloric value of plants wastes

Samples

FW

UPW

EPW

OP

OW

HW

Moisture (M) (wt. %)

11.7

8.2

7.6

5.8

3.4

3.5

Volatile matter (VM), %

71

78.4

63.2

74

76.1

70.5

Fixed carbon, %(FC)

15.5

9.6

24

16.5

18.2

15.4

Ash (dry wt. %)

1.5

2.7

5.1

4.5

2.2

10.4

% Carbon (C)

51

46.9

39.2

54

48.3

46.5

% Oxygen(O)

40.6

42.8

54.2

35.3

45

32

% Hydrogen(H)

7.5

6.7

5.3

7.6

5.9

6.8

% Sulfur(S)

0

0.3

0.09

0.08

0.1

0.4

% Nitrogen(N)

0.2

1.8

1.1

2.3

0.7

3.7

“The characterization of different biomass wastes included proximate analyses, elemental composition, and calorific values, can be seen in Table 5. A high moisture content of 21.5% was obtained, for FW and doesn’t exceed 8.2% for the other samples studied. The ash content is very low for FW in the order of 1.5% and elevated for HW which represents 10.4%. The percentage of volatile matter ranges between 63.2 and 76.1% . FW and OP has a high carbon content with respectively 51 and 54%, and of hydrogen content with respectively 7.5 and 7.6 % compared to other biomass residues. Almost the biomass wastes have a low nitrogen contents and negligible amount of sulfur except HW which contain 3.7% of nitrogen and 0.4% of sulfur”.

Added in the result section.  

Point 2: The caloric value of plant waste must be added.

Response 2:

Thanks for this exciting suggestion, please find the results of the caloric value below, which has been added to Table 5:

Samples

FW

UPW

EPW

OP

OW

HW

Caloric value (MJ/Kg)

19

19.8

14.5

21.6

18.5

12.5

“The caloric value in dry basis of biomass wastes with 18.5-21.6 MJ/kg observed in FW, UPW, OP and OW, indicate a significant energy content in comparison to other biomass residues, while the HW represent a lower percentage of 12.5 MJ/Kg”.

Added in the result section. 

Reviewer 3 Report

Line 40: form of um mixture….correct typo error

Line 46: The production of bioenergy in the world…reframe it

Discuss biomass potential from other countries around the globe in introduction.

Combine section 2.3 and 2.4

Table 1. Correct sexe typo error.

Manuscript need language correction as there are any typo errors.

Author Response

Response to Reviewer 3 Comments

It is a pleasure for our team to receive the recommendations for reviewing our manuscript.

We would like to express our gratitude for the time and expertise you invested in improving our manuscript and enhancing its quality for publication in sustainability journal.

We have carefully considered all suggestions, we have introduced all recommendations, and we have adjusted the manuscript to fit your requirement

We remain available for further comments and adjustments of our manuscript, as well as for additional explanations.

Please find below the itemized answers. 

Point 1: Line 40: form of um mixture….correct typo error

Response 1:

Corrected as follows: form of a mixture”

Changed in abstract line 46

Point 2: Line 46: The production of bioenergy in the world…reframe it

Response 2:

We changed this expression, " The production of bioenergy in the world…" by that:

“Bioenergy production in the world has developed considerably in the last decade”.

Changed in introduction section. line 51.

Point 3: Discuss biomass potential from other countries around the globe in introduction.

Response 3:

Thanks for this kind remark; it is interesting to add the necessary information about the biomass potential in other countries . Accordingly, we added a paragraph of the required information (see below):

“Although this weak production, according to the World Energy Organization, Morocco is considered one of the initiators to invest in bioenergy compared to other north African countries such as Algeria, Tunisia, Libya, and Mauritania, which are characterized by a general weakness in the production of renewable energies and a lack of bioenergy. On the other hand, if we compare Morocco  with other countries on the continent, such as Ethiopia, which produces 290 MW and South Africa 265 MW; it needs more efforts in this field”

Added in introduction section

Point 4: Combine section 2.3 and 2.4

Response 4:

We have combined the section as you recommend as follows :

2.3.        Dosage of sugars and total phenolic compounds

The concentration of reducing sugars is determined by the dinitrosalicylic acid method (DNS) [21]. Briefly, in a 20 ml test tube containing 3 ml of Miller's reagent, 2 ml of test solution is added. After heating for 15 minutes at 100°C and cooling to room temperature, the absorbance is measured at 640 nm.

Total sugars are measured by the Phenol-Sulfuric Acid method by measuring the optical density at 490 nm [22].

Total phenolic compounds were analyzed by the Folin-Ciocalteau method using gallic acid as a standard, as described by Singleton and Rossi [23]

Added in materials and methods section.

Point 5: Table 1. Correct sexe typo error.

Response 5:

sexe typo error was corrected.

Point 6: Manuscript need language correction as there are any typo errors.

Response 6:

Thank for your comment. Indeed, The manuscript was reviewed by an Native English Speaker.

Reviewer 4 Report

 English should improve by a native person. The paper suffers from a poor English structure throughout and cannot be published or reviewed properly in the current format. The manuscript requires a thorough proofread by a native person whose first language is English. The instances of the problem are numerous and this reviewer cannot individually mention them. It is the responsibility of the author(s) to present their work in an acceptable format. Unless the paper is in a reasonable format, it should not have been submitted.

  1. The novelty of the study needs to be highlighted compare to other similar studies.
  2. Discussion is weak. The discussion needs enhancement with real explanations not only agreements and disagreements. Authors should improve it by the demonstration of biochemical/physiological causes of obtained results. Instead of just justifying results, results should be interpreted, explained to appropriately elaborate inferences. Discussion seems to be poor, didn't give good explanations of the results obtained. I think that it must be really improved. Where possible please discuss potential mechanisms behind your observations. You should also expand the links with prior publications in the area, but try to be careful to not over-reach. For the latter, you should highlight potential areas of future study.

4.    The scientific background of the topic is poor. In "Introduction" and "Discussion", the authors should cite recent references between 2016-2020 from JCR journals .

Fahad S, Hasanuzzaman M, Alam M, Ullah H, Saeed M, Ali Khan I, Adnan M. (Eds.) (2020) Environment, Climate, Plant and Vegetation Growth. Springer Nature Switzerland AG 2020. DOI: https://doi.org/10.1007/978-3-030-49732-3

Fahad, S., Sönmez, O., Saud, S., Wang, D., Wu, C., Adnan, M., Turan, V. (Eds.), 2021a. Plant growth regulators for climate-smart agriculture, First edition. ed, Footprints of climate variability on plant diversity. CRC Press, Boca Raton, FL.

Fahad, S., Sönmez, O., Turan, V., Adnan, M., Saud, S., Wu, C., Wang, D. (Eds.), 2021d. Sustainable soil and land management and climate change, First edition. ed, Footprints of climate variability on plant diversity. CRC Press, Boca Raton.

Author Response

Response to Reviewer 4

It is a pleasure for our team to receive the recommendations for reviewing our manuscript.

We would like to express our gratitude for the time and expertise you invested in improving our manuscript and enhancing its quality for publication in Sustainability journal.

We have carefully considered all suggestions, we have introduced all recommendations, and we have adjusted the manuscript to fit your requirement

We remain available for further comments and adjustments of our manuscript, as well as for additional explanations.

Please find below the itemized answers. 

Point 1: English should improve by a native person. The paper suffers from a poor English structure throughout and cannot be published or reviewed properly in the current format. The manuscript requires a thorough proofread by a native person whose first language is English. The instances of the problem are numerous and this reviewer cannot individually mention them. It is the responsibility of the author(s) to present their work in an acceptable format. Unless the paper is in a reasonable format, it should not have been submitted.

Response 1:

Thank you for your comments; Indeed, with the assistance of your pertinent remarks, we have tried to process all your suggestions. the manuscript is improved and it was reviewed by an Native English Speaker.

Point 2:               The novelty of the study needs to be highlighted compare to other similar studies.

Response 2:

Thank you for this kind remark; Indeed, this study compared to other similar studies, allowed us to reveal the following differences:

  • This study is the first one that has been carried out in the context of the valorization of lignocellulosic biomass waste in the Fez-Meknes region.
  • The present study aims to obtain an idea of the types of solid plant waste rejected by the different Moroccan sectors of activity and therefore their importance in terms of added value.
  • The chemical analyses carried out is to find a link between the no treatment of wastes by the industries and the quality of the rejected waste.
  • Highlighting the energetic importance of lignocellulosic biomasses rejected on the horizon to establish a public dumping area for these wastes.

Point 3:               Discussion is weak. The discussion needs enhancement with real explanations not only agreements and disagreements. Authors should improve it by the demonstration of biochemical/physiological causes of obtained results. Instead of just justifying results, results should be interpreted, explained to appropriately elaborate inferences. Discussion seems to be poor, didn't give good explanations of the results obtained. I think that it must be really improved. Where possible please discuss potential mechanisms behind your observations. You should also expand the links with prior publications in the area, but try to be careful to not over-reach. For the latter, you should highlight potential areas of future study.

Response 3:

Thanks for this kind remark; it is interesting to add the necessary explaination about this result. Accordingly, we added two paragraph (see below):

“The high content of cellulose and hemicellulose in most of the analyzed wastes, does not mean that they are easily exploitable. For the reason that the lignocellulosic biomass usually shows low biogas production and biodegradability, due to the recalcitrance of the lignocellulosic complex [28,29]. Thus, the negative correlation between phenolic compounds, total sugars and reducing sugars (Table 3), confirms the inhibitory effect of phenolic compounds on the hydrolysis of polysaccharides into fermentable simple sugars”

“In fact, the problem of waste treatment by production units, according to the survey, cannot be attributed only to poor capacity and lower profit. However also, to ignorance of their energy and industrial values. It became clear through the results of chemical analysis that they are raw materials rich in fibers, which can be burned, transformed into a fuel gas through combustion, into a biogas through fermentation, or into a biomaterial. Therefore, this ignorance of the value of plant waste can be explained by the weak level of training of the owners and managers of these production units, which do not exceed the primary level and, thus, are unable to keep pace with progress in this context”.

Added in discussion section

About the language, the manuscript was reviewed by an English editing service.

Point 4: The scientific background of the topic is poor. In "Introduction" and "Discussion", the authors should cite recent references between 2016-2020 from JCR journals .

Fahad S, Hasanuzzaman M, Alam M, Ullah H, Saeed M, Ali Khan I, Adnan M. (Eds.) (2020) Environment, Climate, Plant and Vegetation Growth. Springer Nature Switzerland AG 2020. DOI: https://doi.org/10.1007/978-3-030-49732-3

Fahad, S., Sönmez, O., Saud, S., Wang, D., Wu, C., Adnan, M., Turan, V. (Eds.), 2021a. Plant growth regulators for climate-smart agriculture, First edition. ed, Footprints of climate variability on plant diversity. CRC Press, Boca Raton, FL.

Fahad, S., Sönmez, O., Turan, V., Adnan, M., Saud, S., Wu, C., Wang, D. (Eds.), 2021d. Sustainable soil and land management and climate change, First edition. ed, Footprints of climate variability on plant diversity. CRC Press, Boca Raton.

Response 4:

Thanks for this exciting suggestion; We added the following references in introduction section:

Fahad S, Hasanuzzaman M, Alam M, Ullah H, Saeed M, Ali Khan I, Adnan M. (Eds.) (2020) Environment, Climate, Plant and Vegetation Growth. Springer Nature Switzerland AG 2020. DOI: https://doi.org/10.1007/978-3-030-49732-3

Fahad, S., Sönmez, O., Turan, V., Adnan, M., Saud, S., Wu, C., Wang, D. (Eds.), 2021d. Sustainable soil and land management and climate change, First edition. ed, Footprints of climate variability on plant diversity. CRC Press, Boca Raton.

Also, we added a paragraph to compare the biomass potentiel in Morocco with others countries in africa as follows:

“Although this weak production, according to the World Energy Organization, Morocco is considered one of the initiators to invest in bioenergy compared to other north African countries such as Algeria, Tunisia, Libya and Mauritania, which are characterized by a general weakness in the production of renewable energies and a lack of bioenergy. On the other hand, if we compare it with other countries on the continent, such as Ethiopia, which produces 290 MW and South Africa 265 MW; it needs more efforts in this field.”

Round 2

Reviewer 1 Report

The author investigated the origin of vegetable waste and analysed the chemical components of the waste. The authors reviesed the manuscript and try to address all my concern. 

Reviewer 2 Report

The author made appropriate modifications to the manuscript.

Reviewer 4 Report

Accept as it stands